**Data Availability Statement:** All relevant data are within the paper and its Supporting information file.

**Funding:** This work was supported by a research grant by Ziemer Ophthalmic System AG, Port,

# Intraoperative complications of cataract surgery using a low-energy femtosecond laser: Results from a real-world high-volume setting

**Julia Riemey** 🔴 *, **Catharina Latz, Alireza Mirshahi**

Dardenne Eye Hospital, Bonn, Germany

* riemey@dardenne.de

## Abstract

In this study, we report data on intraoperative complications occurring after cataract surgery in a high-volume single-center setting using a low-energy, mobile femtosecond laser. We retrospectively reviewed the medical records of patients who underwent femtosecond laser-assisted cataract surgery (FLACS) in our hospital between August 2015 and December 2019. Among the sample of 1,806 eyes of 1,131 patients (903 left and 903 right eyes), the mean age was 75.8 years (range, 21–99 years). The overall intraoperative complication rate was 0.28% (n = 5), with three cases of anterior capsule tear (0.17%) and two cases of posterior capsule tear (0.11%). No further complications occurred. This study underlines the safety of low-energy femtosecond-assisted cataract surgery in a real-world setting with a very low rate of intraoperative complications.

## Introduction

Cataract surgery is one of the most frequently performed procedures in the world [1, 2]. A round, well-centered, and reproducible capsulotomy is essential for successful cataract surgery and intraocular lens (IOL) implantation [1]. The femtosecond laser-assisted technique is special and unique because it allows tissue inside the eye to be cut with very high precision [3]. Femtosecond laser-assisted cataract surgery (FLACS) has enabled cataract surgeons to eliminate the inherent imprecision of manual anterior capsulorhexis techniques and has therefore become a useful tool in cataract surgery in the past decade [1, 4]. As compared with manual capsulorhexis, capsulotomies performed with a femtosecond laser are expected to have less variation in centration and size with reproducible, uniform circular and accurate diameters [1]. Precise, safe, and reproducible capsulotomy is a prerequisite for successful cataract surgery and IOL implantation. Capsulotomies that are performed with a femtosecond laser lead to a more effective lens positioning with reduced probability of IOL tilt and decentration [4].

The latest improvement in femtosecond lasers is the use of low energy. Low-energy technology is a particularly gentle method: while the cutting process is driven by mechanical forces with the high-pulse-energy laser, the tissue can be effectively separated without the need for

Switzerland (https://www.ziemergroup.com/en/contact/addresses-and-locations/ziemer-ophthalmic-systems-ag/) The funders had no role in study design, data collection and analysis and decision to publish. The funders reviewed manuscript.

**Competing interests:** We would also like to make a declaration of competing interests. AM is a consultant to Ziemer Ophthalmics, Port, Switzerland. This does not alter our adherence to PLOS ONE policies on sharing data and materials.

secondary mechanical tearing effects [3]. Therefore, using low pulse energy results in a very smooth surface, and the adjacent tissue remains virtually undamaged [3]. A high-pulse-energy uses larger spacing between spots, whereas a low-pulse-energy laser allows creation of a larger number of overlapping smaller spots [3]. Recent changes in the numerical aperture of laser-focusing optics and the repetition rate of laser sources have further decreased collateral damage while increasing precision. By enlarging the numerical aperture of the focusing optics, the pulse energy threshold for optical breakdown decreases, resulting in cutting with almost little collateral damage.

Publications reporting the results of low-energy lasers are still very sparse. In this study, we report data on the intraoperative complications of cataract surgery using a low-energy femtosecond laser in a high-volume single-center setting in Bonn, Germany. in this study, we examine the intraoperative complication rate of FLACS performed by one of two experienced surgeons.

## Materials and methods

This was a retrospective, single-center, consecutive case series. We reviewed the medical records of patients who underwent low-energy FLACS with Ziemer LDV Z8 (Ziemer Ophthalmic Systems AG, Port, Switzerland) between August 1, 2015, and December 31, 2019. Unlike most femtosecond lasers, the Ziemer FEMTO LDV Z8 uses low pulse energy (nJ) and a high repetition rate in the MHz range. All patients underwent an operation at the Dardenne Eye Hospital (Bonn, Germany), which was performed by two experienced high-volume cataract surgeons (A.M., K.T.). The first 20 operations performed by each surgeon were considered learning experience and as such were excluded from the analysis. We reviewed the surgery reports with special attention paid to the occurrence of intraoperative complications, including but not limited to capsular complications during FLACS. An intraoperative complication existed when the surgery report was marked as "impeded with complications." In case of a complication, we collected further data concerning patient demographics, additional ocular diagnoses, ocular biometry, and the outcome. Observational data were calculated using Excel (Microsoft Office Professional Plus 2013). We calculated the complication rate and confidence limits using the RStudio 1.4.1717 two-sided exact binomial test, assuming that the complication rate was not zero. This study was performed according to the tenets of the Declaration of Helsinki and did not require approval of an independent ethics committee, as ruled by the North Rhine Medical Chamber, due to the retrospective and epidemiological design as stated in §15 of the code of medical ethics by the North Rhine Chamber of Physicians. All included patients were aware of the use of their data for research purposes. Informed consent was waived.

### Femtosecond laser

Femtosecond lasers are a more recent advance in solid-state laser technology. These lasers operate at near-infrared wavelengths at pulse durations of less than 1 picosecond (ps). Because the threshold radiant exposure ($J/cm^2$) for inducing optical breakdown in tissue is about two orders of magnitude lower in the femtosecond (fs) pulse duration regime than at 10 nanoseconds (ns) [5], much lower pulse energies can be applied to separate tissue. High pulse repetition rates from 10 s of kHz to even MHz are then used to create continuous cut planes inside the tissue, by placing many pulses close to each other with the help of three-dimensional focus scanner beam scanning systems.

Lower pulse energies lead to a drastic reduction in the mechanical side effects of optical breakdown. For 300-fs pulses of 0.75-µJ energy, the generated cavitation bubbles have radii of only 45 µm, almost two orders of magnitude smaller than ns pulse with energies in the mJ range [6]. In addition, the associated pressure waves are much weaker, 1–5 bar at 1 mm

distance [7]. This process is referred to as "plasma-induced ablation," as the disruptive mechanical side effects of ns pulses are basically absent. In addition, the thermal side effects of fs pulses in tissue are almost negligible [8].

The pulse energy required to achieve optical breakdown can be reduced in two ways. First, by shortening the pulse duration—the latest fs lasers can achieve pulse durations of 200–300 fs, whereas earlier models had pulse durations of up to 800 fs. Second, by reducing the focal spot size and the beam waist. In other words, the focal volume varies inversely with the cube of the numerical aperture of the focusing optics. The larger the numerical aperture, the smaller the focal spot and, finally, the smaller the energy threshold for optical breakdown [9].

## Surgical technique

Almost all surgeries were performed under peribulbar anesthesia (4 mL bupivacaine 0.75% and 2 mL mepivacaine 2% + 75 IE hyaluronic acid). In cases with increased risk of bleeding (i.e., continuous anticoagulation), the surgery was performed under topical anesthesia. Following disinfection, the eye was draped in a sterile manner. A Lieberman speculum was used, and the Ziemer FEMTO LDV Z8 suction ring was applied to the eye. If needed, the speculum was further widened or placed at an angle to allow for optimized suction. Once a vacuum was achieved, the patient interface was securely attached to the eye. Next, 2–5 mL of balanced salt solution (BSS) was applied to the patient interface, and the femtosecond handpiece was docked. Attention was then focused on the femtosecond laser monitor. Capsulotomy size and position, as well as lens fragmentation location, were suggested automatically by the laser or adjusted according to the patient's individual optical coherence tomography (OCT) images, if needed. The standard preset was a 5.2-mm-diameter capsulotomy centered on the pupil with 95% laser energy and nuclear fragmentation with a 6-mm diameter in an eight-segment pattern with 110% laser energy. After capsulotomy and fragmentation were complete, the vacuum was released, and the laser handpiece was removed. Next, a clear corneal or near-clear corneal tunnel of 2.4-mm width and approximately 2.5-mm length was placed at either the 12 o'clock or temporal position. A dispersive viscoelastic was used to fill the anterior chamber, and two paracenteses of 0.9 mm each were placed. Next, the integrity of the capsulotomy was controlled with Utrata forceps, and the precut anterior capsule was removed from the anterior chamber. To mobilize the nucleus, gentle hydrodissection and hydrodelination were performed. The 2.4-mm phacotip (Centurion, Alcon, Freiburg, Germany) was introduced through the corneal tunnel, and a chopper was introduced through the side port. Nuclear disassembly was performed by mechanically separating and aspirating the precut pieces. The epinucleus was rotated and removed using low phaco and moderate vacuum settings. The cortex was aspirated using the bimanual irrigation/aspiration (I/A) system. The capsule was polished by jet irrigation with BSS. The IOL was implanted into the capsular bag either with viscoelastic stabilization or, if the IOL was shootable, under irrigation. The IOL was centered, and the optimal position of the optic and haptics was controlled. The remaining viscoelastic was removed with the I/A handpieces. The wounds were hydrated and checked for water tightness. 1 mg cefuroxim was administered intracamerally in addition to subconjunctival dexamethasone 1 mL (4 mg/mL), as well as subtenonal triamcinolone 1 mL (40 mg/mL). The speculum was removed, and ofloxacin ointment was introduced to the fornix. The eye was dressed with an eye bandage.

## Results

We reviewed the records of 1,806 eyes of 1,131 patients (Table 1). A total of 903 left eyes and 903 right eyes were operated by either of two experienced cataract surgeons. The patients were on average 75.7 years old, with a minimum age of 21 years and a maximum age of 99 years.

**Table 1. Overview of the patient sample.**

|  | Sample | Complication cases |
|---|---|---|
| Patients | 1,131 | 5 |
| Eyes | 1,806 | 5 |
| Right eye | 903 | 1 |
| Left eye | 903 | 4 |
| Mean age, years | 75.75 | 78.00 |
| Age, years (minimum) | 21 | 56 |
| Age, years (maximum) | 99 | 89 |

Among the study sample, intraoperative complications occurred in five eyes of five patients, which correspond to a complication rate of 0.28% (95% confidence interval 0.09%–0.65%, $p < 0.05$; see Table 2). All complications were capsule related: three eyes had an anterior capsular tear (0.17%), whereas the other two eyes had a posterior capsule rupture (0.11%). We did not experience any suction losses. IOL implantation was successful in all cases (100%). In three of the five cases (60%), the IOL could be implanted into the capsular bag. Sulcus implantation was performed in two cases (40%) with posterior capsular tears. Anterior vitrectomy was required in one case of posterior capsular rupture.

For the sake of completeness, we also present the number of complications considering the first 20 surgeries of each surgeon. An additional two complications were observed among the initial 20 cases of each surgeon, for a total of seven complications reported among 1,846 of the operated eyes. The subsequent complications occurred in surgery numbers 106, 383, 476, 891, and 1,416.

Patients with complications were on average 78.00 years old. The youngest patient experiencing a complication was 56 years old, while the oldest person with a complication was 89 years old. Four complications occurred when operating the right eyes; one complication appeared in the left eye. Each patient with a complication had at least one diagnosis aggravating surgery: hard nucleus (3), posterior subcapsular cataract (1), primary posterior capsular fibrosis (1), floppy iris (1), narrow pupil (1), flat anterior chamber (1), and positive posterior pressure (2) (see Table 3). None of the patients suffered from diabetes or were taking prostate medication.

## Discussion

The results of our study underline the safety of low-energy FLACS in a high-volume, real-world setting. The only observed complications were capsule related, and those occurred at a very low incidence rate of 0.28%. Among many factors that could influence the complications rates in cataract surgery, two should be emphasized: (1) both surgeons who performed the surgeries in this study were high-volume surgeons (>15,000 cases in total and >1,000 cases per year), potentially resulting in a lower intraoperative complication rate as compared

**Table 2. Overview of complication cases.**

|  | n (%) |
|---|---|
| Complication | 5 (0.28) |
| Anterior capsular tear | 3 (0.17) |
| Posterior capsule rupture | 2 (0.11) |
| Successful implantation of IOL | 5 (100) |
| In the capsular bag | 3 (60) |
| In the sulcus | 2 (40) |

**Table 3. Diagnoses aggravating surgery in patients with an intraoperative complication.**

|  | Patient | | | | | |
| --- | --- | --- | --- | --- | --- | --- |
| **Complication** | **1** | **2** | **3** | **4** | **5** | **Sum** |
| Hard nucleus |  | x |  | x | x | 3 |
| Posterior subcapsular opacity |  |  |  | x |  | 1 |
| Primary posterior capsular Fibrosis |  |  |  | x |  | 1 |
| Floppy iris | x |  |  |  |  | 1 |
| Narrow pupil | x |  |  |  |  | 1 |
| Flat anterior chamber |  | x |  |  |  | 1 |
| Positive posterior pressure |  |  | x |  | x | 2 |
| Sum | 2 | 2 | 1 | 3 | 2 |  |

with nonexperienced surgeons, and (2) the use of a low-energy femtosecond laser proved to be safe in our case series.

## Intraoperative complications in low- and high-energy FLACS

Numerous studies have reported the results of FLACS with high-energy femtosecond laser. In a large meta-analysis, Popovic et al. [10] observed an overall incidence of complications of 379 events in a total of 3,704 eyes, corresponding to a complication rate of 10.23%.

The high-energy method generates more stress or potentially even damage to the adjacent tissue [3]. Compared with the high-energy concept, the low-energy method generates a uniquely smooth surface, which hardly damages the adjacent tissue at all [3]. This results in lower amounts of intraoperative prostaglandin release and, thereby, no or negligible intraoperative pupil narrowing [11]. An increased level of prostaglandin may be responsible for the intraoperative narrowing of the pupil [11]. Findings of Schwarzenbacher et al. [12] underline that the low-energy concept produces a negligible inflammatory response: inflammatory cytokines interleukin (IL)-1b and IL-6 were not elevated after pretreatment with low-energy femtosecond laser [12]. In another study including 52 eyes that had undergone FLACS, no statistically significant changes were detected in the pupil area when comparing preoperative pupil status with postlaser size [11].

Literature on the use of the low-energy laser is still very sparse. Recently, Lin et al. [13] reported comparative results of a low- versus high-energy FLACS in 200 eyes of 200 patients. While the incision completeness and side incisions were comparable between the two groups, the low-energy device delivered better integrity of capsulotomies, with less intraoperative pupil narrowing [13]. Existing studies have all reported no major intraoperative complications using the low-energy laser [14–17]. In a study with 14 eyes undergoing high-frequency, low-energy FLACS, no major complications, such as anterior capsule tears or posterior capsule ruptures, occurred [16]. In comparing low-energy FLACS with standard phacoemulsification, Cavallini et al. [14] did not observe any major intraoperative complications. In the low-energy group consisting of 70 eyes, the incision was incomplete in two cases; no incomplete capsulotomies were reported [14]. These findings are congruent with our results: based on a much larger sample size (1,806 eyes), we observed a comparatively low complication rate. Hence, one side effect of lower "collateral damage" when using the low-energy method may be less complicated surgeries.

## Capsule complications in FLACS

Because we observed only capsule complications in our study, it is worthwhile to take a closer look at this specific type of complication with regard to FLACS.

In the beginning of FLACS technology, the integrity of laser anterior capsulotomy was reported to be compromised as compared with phacoemulsification capsulotomy, because of aberrant pits that created postage-stamp perforations [18]. The quality of the laser capsulotomy has been improved since then: the smoothness of the capsulotomy edge was improved by decreasing the spot size and separation. In addition, the overall time it takes to complete a laser capsulotomy and nuclear fragmentation has been reduced. Moreover, the impact of the laser energy not only on the capsulotomy but also on the epithelial cells of the lens has been further evaluated [19–21]. Pulse energies of 13 μJ and higher showed significantly more inflammation and apoptosis in lens epithelial cells when compared with either manual rhexis or pulse energies of 10 μJ or less [19, 20]. With the low-energy, high-frequency system, pulse energies range from $10^3$ lower. Here when different energy levels of 90%, 130%, and 150% were compared, apoptosis was not increased in a dose-dependent manner [22]. In addition, capsulotomy strength was not significantly different across the three different energy settings [22]. Thus, a very low rate of capsule tears has been reported for the Ziemer FEMTO LDV Z8 (see Table 4). In 2020, Kolb et al. [17] published a meta-analysis and systematic review in which 218 eyes were treated with the low-energy Ziemer FEMTO LDV Z8, and no anterior or posterior capsular tears occurred [17]. With the high-energy laser, 78 events of anterior capsular tear occurred in 7,804 eyes (1.00%) and 30 events of posterior capsule rupture occurred in 6,973 eyes (0.43%) [17]. Wang et al. [23] and Popovic et al. [10] also reported a capsule complication rate of 2.35% and 5.27%, respectively, for high-energy FLACS, whereas Pajic et al. [15] reported no major capsular complications with the use of a low-energy device.

In accordance with the few reported capsular complications in the low-energy setting, our study included a much larger sample size with similarly low capsule complications. While performing FLACS with the Ziemer FEMTO LDV Z8, we noticed that the integrity of the capsulotomy was substantially higher when the aimed capsulotomy position was moved anteriorly in relation to the OCT-measured capsule position. We hypothesized that the nuclear fragmentation prior to capsulotomy caused an anteriorization of the lens capsule apparatus by creating gas bubbles inside the lens capsule complex.

## FLACS in comparison with phacoemulsification

An insight into the results published in the scientific literature will provide an overview regarding how the operative outcomes of FLACS compare with those of standard phacoemulsification. According to published studies, the two procedures do not differ in terms of visual and refractive outcomes or in overall [10] or intra- and postoperative complication rates [23].

**Table 4. Capsule complication rates by femtosecond laser device in different publications.**

|  | Femtosecond laser device | Posterior capsule complication rate | Anterior capsule complication rate | Capsule complication rate |
|---|---|---|---|---|
| Popovic et al. (2016) | High energy | 0.89% (n = 3,390) | Not reported | 5.27% (n = 3,571) |
| Kolb et al. (2020) | High energy | 0.43% (n = 6,973) | 1.00% (n = 7,804) | 1.38% (n = 7,837) |
| Wang et al. (2019) | High energy | 0.62% (n = 3,080) | 1.77% (n = 3,113) | 2.35% (n = 3,153) |
| Pajic et al. (2017) | Low energy | Not reported | Not reported | 0.00% (n = 68) |
| Kolb et al. (2020) | Low energy | 0.00% (n = 218) | 0.00% (n = 218) | 0.00% (n = 218) |
| Our study | Low energy | 0.11% (n = 1,806) | 0.17% (n = 1,806) | 0.28% (n = 1,806) |

However, Chen et al. [24] reported a statistically significant lower rate of intraoperative complications (1.8%) in FLACS subjects in comparison with subjects undergoing the traditional phacoemulsification technique (5.8%), investigating 273 eyes and 553, respectively. Complication rates using the high-energy laser device ranged from 0% in three surgeons; to 1.8% and 2.4% in another two surgeons, respectively; and up to 5.8% in one surgeon. Four of five surgeons in the study had a lower complication rate when conducting the laser-assisted procedure. An open posterior capsule occurred two times in FLACS, whereas it occurred 10 times in traditional phacoemulsification [24]. For FLACS, this corresponds to a posterior capsule complication rate of 0.73. Referring to this result, in our study posterior capsule complication rate was lower, at 0.11%.

In comparing low-energy FLACS using Ziemer FEMTO LDV Z8 with bimanual phacoemulsification, Cavallini et al. [14] did not observe any intraoperative complications, either in the phacoemulsification group or in the group with the low-energy Ziemer FEMTO LDV Z8.

Based on the results of our study, capsular tears may be expected in 0.28% of cases. These findings underline that the use of this method does not cause many intraoperative complications and is accordingly in line with results reported in the literature [3, 23, 25]. Our results emphasize that an accurate and circular capsulotomy has low risk for an intraoperative complication.

## Learning curve

In reviewing the literature, we found that the use of the FLACS technique entails a learning curve [26]. Several studies have indicated increased safety after a learning curve [27–29]. Roberts et al. [28] confirmed that the incidence of anterior and posterior capsular rupture decreased significantly with the number of performed surgeries, decreasing from 7.5% in the first 200 cases to 0.62% in the last 1,300 cases. Cavallini et al. [29] also observed an initial learning curve: for the first 60 FLACS, the intraoperative complication rate was 18.3%, whereas it was 3.3% for the following 60 FLACS cases. Nagy et al. [30] showed that most complications in FLACS occurred principally during the first 100 surgeries. We took this learning curve into account by excluding the first 20 operations of each surgeon in our study. Nevertheless, even if the first 20 operations of the two surgeons were included in our study, the complication rate would hardly change: 0.28% (5/1,806)–0.38% (7/1,846). It should not go unmentioned that an exclusion of up to 100 cases would not have changed the number of complications in our study. The first complication after the learning curve occurred in surgery 106. This indicates a potentially shorter learning curve with the low-energy laser as opposed to the high-energy technology.

## Conclusions

FLACS allows tissue to be cut with a very high precision inside the eye. In our study, we were able to demonstrate a low intraoperative complication rate of 0.28% in a high-volume single-surgical center setting using a low-energy femtosecond laser in the hands of two experienced cataract surgeons. Specifically, anterior capsule tears occurred in 0.17% of cases and posterior capsule ruptures in 0.11% of cases. We believe that this is related to the low-energy technology, which results in virtually no collateral damage adjacent to the laser spots and thereby improves precision and safety of the procedure. However, this hypothesis must be validated by further investigations, especially on comparative complication data between femtosecond laser-assisted and conventional cataract surgery in the hands of the same surgeon. Despite this, insights must be validated in an even larger sample size with a matched control group. More studies comparing high- with low-energy concepts are needed.

## Supporting information

**S1 File.**
(XLSX)

## Author Contributions

**Conceptualization:** Alireza Mirshahi.

**Data curation:** Julia Riemey.

**Formal analysis:** Julia Riemey.

**Funding acquisition:** Julia Riemey, Alireza Mirshahi.

**Methodology:** Alireza Mirshahi.

**Project administration:** Julia Riemey.

**Visualization:** Julia Riemey.

**Writing – original draft:** Julia Riemey, Catharina Latz.

**Writing – review & editing:** Catharina Latz, Alireza Mirshahi.

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
