## [Decision Letter · Decision Letter 0]

8 Aug 2022

PONE-D-22-10143Intraoperative complications of cataract surgery using a low-energy femtosecond laser: Results from a real-world high-volume settingPLOS ONE

Dear Dr.Julia Riemey:

Thank you for submitting your manuscript to PLOS ONE. After careful consideration, we feel that it has merit but does not fully meet PLOS ONE’s publication criteria as it currently stands. Therefore, we invite you to submit a revised version of the manuscript that addresses the points raised during the review process.

 Reviewers' comments:

Reviewer's Responses to Questions

**Comments to the Author**

1. Is the manuscript technically sound, and do the data support the conclusions?

Reviewer #1: Yes

Reviewer #2: Yes

2. Has the statistical analysis been performed appropriately and rigorously? 

Reviewer #1: Yes

Reviewer #2: Yes

3. Have the authors made all data underlying the findings in their manuscript fully available?

Reviewer #1: Yes

Reviewer #2: Yes

4. Is the manuscript presented in an intelligible fashion and written in standard English?

Reviewer #1: No

Reviewer #2: Yes

5. Review Comments to the Author

Reviewer #1: 1. Abbreviations used in the manuscript has not been mentioned with their relevant word or phrase

2. The manuscript is written with poor grammar and spelling mistakes which minimizes the overall quality.

Reviewer #2: The study underlines the safety of low-energy femtosecond assisted cataract surgery in a real-world setting. The manuscript is well written , designed and analyzed. The concept of cataract surgery using low-energy lasers are still very sparse. This study reported data on intraoperative complications of cataract surgery using a low-energy

46 femtosecond laser in a high-volume single-center setting in Bonn, Germany.

6. PLOS authors have the option to publish the peer review history of their article (what does this mean?). If published, this will include your full peer review and any attached files.

Reviewer #1: No

Reviewer #2: No

We look forward to receiving your revised manuscript.

Kind regards,

Rajiv Janardhanan, Ph.D.

Academic Editor

PLOS ONE

Journal Requirements:

I have read the journal's policy and the authors of this manuscript have the following competing interests: AM is a consultant to Ziemer Ophthalmics, Port, Switzerland.

---

## [Author Response · Author response to Decision Letter 0]

29 Sep 2022

Dear Reviewer

thank you very much for your effort and review of this paper. Please find enclosed our revised manuscript titled “Intraoperative complications of cataract surgery using a low-energy femtosecond laser: Results from a real-world high-volume setting”.

Following the suggestions of Reviewer #2, we have developed and expanded the concept of cataract surgery using low-energy lasers. The low-energy-concept is now carried out in detail.

Following the comments of Reviewer #1, we carefully reworked our abbreviations used in the manuscript. Mentioned abbreviations are now explained with their first use.

The revised manuscript has subsequently been carefully reviewed by an experienced editor whose first language is English and who specializes in editing papers written by scientists whose native language is not English.

We look forward to hearing from you at your earliest convenience.

Sincerely,

Julia Riemey

---

## [Decision Letter · Decision Letter 1]

25 Oct 2022

PONE-D-22-10143R1Intraoperative complications of cataract surgery using a low-energy femtosecond laser: Results from a real-world high-volume settingPLOS ONE

Dear Dr. Julia Riemey:

Thank you for submitting your manuscript to PLOS ONE. After careful consideration, we feel that it has merit but does not fully meet PLOS ONE’s publication criteria as it currently stands. Therefore, we invite you to submit a revised version of the manuscript that addresses the points raised during the review process.

Reviewer #1: The manuscript has been well written but the reference number are not in the order. Kindly arrange the references in the manuscript in the systematic fashion.

Please submit your revised manuscript by 02 November 2022. If you will need more time than this to complete your revisions, please reply to this message or contact the journal office at plosone@plos.org. Please include the following items when submitting your revised manuscript:A rebuttal letter that responds to each point raised by the academic editor and reviewer(s). You should upload this letter as a separate file labeled 'Response to Reviewers'.A marked-up copy of your manuscript that highlights changes made to the original version. You should upload this as a separate file labeled 'Revised Manuscript with Track Changes'.An unmarked version of your revised paper without tracked changes. You should upload this as a separate file labeled 'Manuscript'.If applicable, we recommend that you deposit your laboratory protocols in protocols.io to enhance the reproducibility of your results. Protocols.io assigns your protocol its own identifier (DOI) so that it can be cited independently in the future. For instructions see: https://journals.plos.org/plosone/s/submission-guidelines#loc-laboratory-protocols. Additionally, PLOS ONE offers an option for publishing peer-reviewed Lab Protocol articles, which describe protocols hosted on protocols.io. Read more information on sharing protocols at https://plos.org/protocols?utm_medium=editorial-email&utm_source=authorletters&utm_campaign=protocols.

We look forward to receiving your revised manuscript.

Kind regards,

Rajiv Janardhanan, Ph.D.

Academic Editor

PLOS ONE

Journal Requirements:

Reviewers' comments:

Reviewer's Responses to Questions

**Comments to the Author**

1. If the authors have adequately addressed your comments raised in a previous round of review and you feel that this manuscript is now acceptable for publication, you may indicate that here to bypass the “Comments to the Author” section, enter your conflict of interest statement in the “Confidential to Editor” section, and submit your "Accept" recommendation.

Reviewer #1: All comments have been addressed

2. Is the manuscript technically sound, and do the data support the conclusions?

Reviewer #1: Yes

3. Has the statistical analysis been performed appropriately and rigorously? 

Reviewer #1: Yes

4. Have the authors made all data underlying the findings in their manuscript fully available?

Reviewer #1: Yes

5. Is the manuscript presented in an intelligible fashion and written in standard English?

Reviewer #1: Yes

6. Review Comments to the Author

Reviewer #1: The manuscript has been well written but the reference number are not in the order. Kindly arrange the references in the manuscript in the systematic fashion.

7. PLOS authors have the option to publish the peer review history of their article (what does this mean?). If published, this will include your full peer review and any attached files.

Reviewer #1: No

---

## [Author Response · Author response to Decision Letter 1]

1 Nov 2022

Dear Reviewer

thank you very much for your effort and review of this paper. Please find enclosed our revised manuscript titled “Intraoperative complications of cataract surgery using a low-energy femtosecond laser: Results from a real-world high-volume setting”.

Following the comments of Reviewer #1, we arranged the references in the manuscript in the systematic fashion. The reference numbers are now in the right order.

We look forward to hearing from you at your earliest convenience.

Sincerely,

Julia Riemey

---

## [Decision Letter · Decision Letter 2]

29 Nov 2022

Intraoperative complications of cataract surgery using a low-energy femtosecond laser: Results from a real-world high-volume setting

PONE-D-22-10143R2

Dear Dr.HOUDA ZAHFIR

We’re pleased to inform you that your manuscript has been judged scientifically suitable for publication and will be formally accepted for publication once it meets all outstanding technical requirements.

Kind regards,

Rajiv Janardhanan, Ph.D.

Academic Editor

PLOS ONE

Reviewers' comments:

Reviewer's Responses to Questions

**Comments to the Author**

1. If the authors have adequately addressed your comments raised in a previous round of review and you feel that this manuscript is now acceptable for publication, you may indicate that here to bypass the “Comments to the Author” section, enter your conflict of interest statement in the “Confidential to Editor” section, and submit your "Accept" recommendation.

Reviewer #1: All comments have been addressed

2. Is the manuscript technically sound, and do the data support the conclusions?

Reviewer #1: Yes

3. Has the statistical analysis been performed appropriately and rigorously? 

Reviewer #1: Yes

4. Have the authors made all data underlying the findings in their manuscript fully available?

Reviewer #1: Yes

5. Is the manuscript presented in an intelligible fashion and written in standard English?

Reviewer #1: Yes

6. Review Comments to the Author

Reviewer #1: (No Response)

7. PLOS authors have the option to publish the peer review history of their article (what does this mean?). If published, this will include your full peer review and any attached files.

Reviewer #1: No

---

## [Editor Report · Acceptance letter]

6 Dec 2022

PONE-D-22-10143R2 

Intraoperative complications of cataract surgery using a low-energy femtosecond laser: Results from a real-world high-volume setting 

Dear Dr. Riemey:

I'm pleased to inform you that your manuscript has been deemed suitable for publication in PLOS ONE. Congratulations! Your manuscript is now with our production department. 

Kind regards, 

on behalf of

Dr. Rajiv Janardhanan 

Academic Editor

PLOS ONE